



# Surface and subsurface characterisation of salt pans expressing polygonal patterns

Jana Lasser[1,2], Joanna M. Nield[3], and Lucas Goehring[4]

[1]Max Planck Institute for Dynamics and Self-Organization, Am Fassberg 17, 37077 Göttingen, Germany
[2]Complexity Science Hub Vienna, Josefstädterstrasse 39, 1080 Wien
[3]Geography and Environmental Science, University of Southampton, Highfield, Southampton SO17 1BJ, UK
[4]School of Science and Technology, Nottingham Trent University, Nottingham NG11 8NS, UK

**Correspondence:** Jana Lasser (lasser@csh.ac.at)

**Abstract.** The data set described here contains information about the surface, subsurface and environmental conditions of salt pans that express polygonal patterns in their surface salt crust. Information stems from 5 field sites at Badwater Basin and 21 field sites at Owens Lake – both in central California. All data was recorded during two field campaigns, from between November and December, 2016, and in January 2018. Crust surfaces, including the mean diameter and fluctuations in the height of

the polygonal patterns, were characterised by terrestrial laser scanner (Nield et al., 2020b), DOI 10.1594/PANGAEA.911233. The data contains the resulting three dimensional point clouds, which describe these surfaces. The subsurface is characterised by grain size distributions of samples taken from depths between 5 cm and 100 cm below the salt crust, and measured with a laser particle size analyser (Lasser and Goehring, 2020b), DOI 10.1594/PANGAEA.910996. Subsurface salinity profiles were recorded and the ground water density was also measured (Lasser and Goehring, 2020a), DOI 10.1594/PANGAEA.911059.

Additionally, the salts present in the crust and pore water were analysed to determine their composition (Lasser and Karius, 2020), DOI 10.1594/PANGAEA.911239. To characterise the environmental conditions at Owens Lake, including the differences between nearby crust features, records were made of the temperature and relative humidity during one week in November 2016 (Nield et al., 2020a), DOI 10.1594/PANGAEA.911139. The field sites are characterised by images (Lasser et al., 2020), DOI 10.1594/PANGAEA.911054, showing the general context of each site, such as pictures of selected salt polygons, includ-

ing any which were sampled, a typical core from each site at which core samples were taken and close-ups of the salt crust morphology. Finally, two videos of salt crust growth over the course of spring 2018 and reconstructed from time-lapse images are included (Lasser et al., 2020), DOI 10.1594/PANGAEA.911054.



## 1 Introduction

Salt pans play an important role in climate-surface-interactions (e.g. Gill (1996); Prospero (2002); Nield et al. (2015)). Occurring around the world, they are often covered by a salt crust expressing polygonal ridge patterns with diameters of roughly one to three meters and ridge heights up to $0.4\,\mathrm{m}$ (e.g. Christiansen (1963); Krinsley (1970); Nield et al. (2015); Lasser et al. (2019)). These iconic patterned surfaces annually draw millions of tourists to sites like Salar de Uyuni or Death Valley (Service, 2019), and some examples are shown in Fig. 1. The salt crusts themselves are dynamic over months to years (Lowenstein and Hardie, 1985; Lokier, 2012; Nield et al., 2013, 2015) and the ridges interact with the often strong winds blowing over the surface. The wind erodes the surface and carries sand and small salt particles into the atmosphere. As such, salt pans are amongst the largest sources of atmospheric dust on the globe (Gill, 1996; Prospero, 2002).

To date, the driving mechanism of the pattern formation in salt crusts is unclear and has been under debate in the literature. Crust patterns have been attributed to buckling or wrinkling as expanding areas of crust collide (Christiansen, 1963; Fryberger et al., 1983; Lowenstein and Hardie, 1985), or to surface cracks (Krinsley, 1970; Dixon, 2009; Tucker, 1981; Deckker, 1988; Lokier, 2012). The data presented in this publication was gathered during research in support of a new hypothesis (Lasser et al., 2019), which attributes the salt polygons visible at the surface to convective flows of saline water in the ground below the patterns. To this end, a characterisation of the surface relief at various sites (Nield et al., 2020b), the general site conditions (Lasser et al., 2020), minerals present in the crusts (Lasser and Karius, 2020)), the subsurface soil composition (Lasser and Goehring, 2020b), the spatial salt distribution below the patterns (Lasser and Goehring, 2020a), groundwater density (Lasser and Goehring, 2020a)) as well as the temperature and relative humidity at various crust features (Nield et al., 2020a) was made. The study methodology and resulting data are detailed in this work. The associated data sets are freely available at the PANGAEA data repository.

## 2 Materials and Methods

### 2.1 Research area

We carried out two field campaigns to salt pans in central California, the first between November and December, 2016, and the second in January, 2018. During the first campaign we conducted a broad survey of several dry lakes in the region. We focused on Owens Lake and Badwater Basin but also briefly visited Soda Lake and Bristol Dry Lake, where we either found no polygons (Soda Lake, near Zzyzx) or a crust that was significantly disturbed by salt mining operations (Bristol Dry Lake, adjacent to Amboy Rd.). During the second field campaign we visited Owens Lake only and focused on surface scans and the collection of samples to compile high resolution subsurface salt concentration profiles. Across both trips we visited a total of 21 sites at Owens Lake and 5 sites at Badwater Basin; site designations and GPS coordinates are indicated in Table 1.



| Location | Label | Latitude | Longitude | Year |
|---|---|---|---|---|
| Death Valley | Badwater P1 | 36°13.651′ | -116°46.723′ | 2016 |
| Death Valley | Badwater P2 | 36°13.674′ | -116°46.735′ | 2016 |
| Death Valley | Badwater P3 | 36°13.665′ | -116°46.820′ | 2016 |
| Death Valley | Badwater P4 | 36°13.660′ | -116°46.903′ | 2016 |
| Death Valley | Badwater P5 | 36°13.654′ | -116°47.036′ | 2016 |
| Owens Lake | T10-3 P1 | 36°23.147′ | -117°56.772′ | 2018 |
| Owens Lake | T16 P1 | 36°23.953′ | -117°56.454′ | 2018 |
| Owens Lake | T2-4 P1 | 36°20.803′ | -117°58.642′ | 2016 |
| Owens Lake | T2-5 P1 | 36°21.055′ | -117°58.824′ | 2016 |
| Owens Lake | T2-5 P2 | 36°20.895′ | -117°58.740′ | 2016 |
| Owens Lake | T2-5 P3 | 36°20.877′ | -117°58.711 | 2018 |
| Owens Lake | T25-3 P1 | 36°27.039′ | -117°54.510′ | 2018 |
| Owens Lake | T25-3 P2 | 36°28.383′ | -117°54.957′ | 2018 |
| Owens Lake | T27-A P1 | 36°29.302′ | -117°55.953′ | 2016 |
| Owens Lake | T27-A P2 | 36°29.061′ | -117°55.602′ | 2016 |
| Owens Lake | T27-A P3 | 36°29.112′ | -117°55.804′ | 2018 |
| Owens Lake | T27-S P1 | 36°28.549′ | -117°54.994′ | 2018 |
| Owens Lake | T29-3 P1 | 36°29.955′ | -117°55.999′ | 2016 |
| Owens Lake | T29-3 P2 | 36°29.960′ | -117°55.962′ | 2016 |
| Owens Lake | T32-1-L1 P1 | 36°53.897′ | -117°57.209′ | 2016 |
| Owens Lake | T32-1-L1 P1 | 36°32.354′ | -117°57.218′ | 2018 |
| Owens Lake | T32-1-L1 P3 | 36°32.337′ | -117°57.204′ | 2018 |
| Owens Lake | T36-3 P1 | 36°29.953′ | -117°58.505′ | 2016 |
| Owens Lake | T36-3 P2 | 36°30.050′ | -117°58.518′ | 2016 |
| Owens Lake | T36-3 P3 | 36°29.724′ | -117°57.916′ | 2016 |
| Owens Lake | T8-W P1 | 36°22.522′ | -117°57.256′ | 2018 |

**Table 1.** Location, site label, GPS coordinates and year of data collection for sites at Badwater Basin (Death Valley, CA) and Owens Lake (Owens Valley, CA).

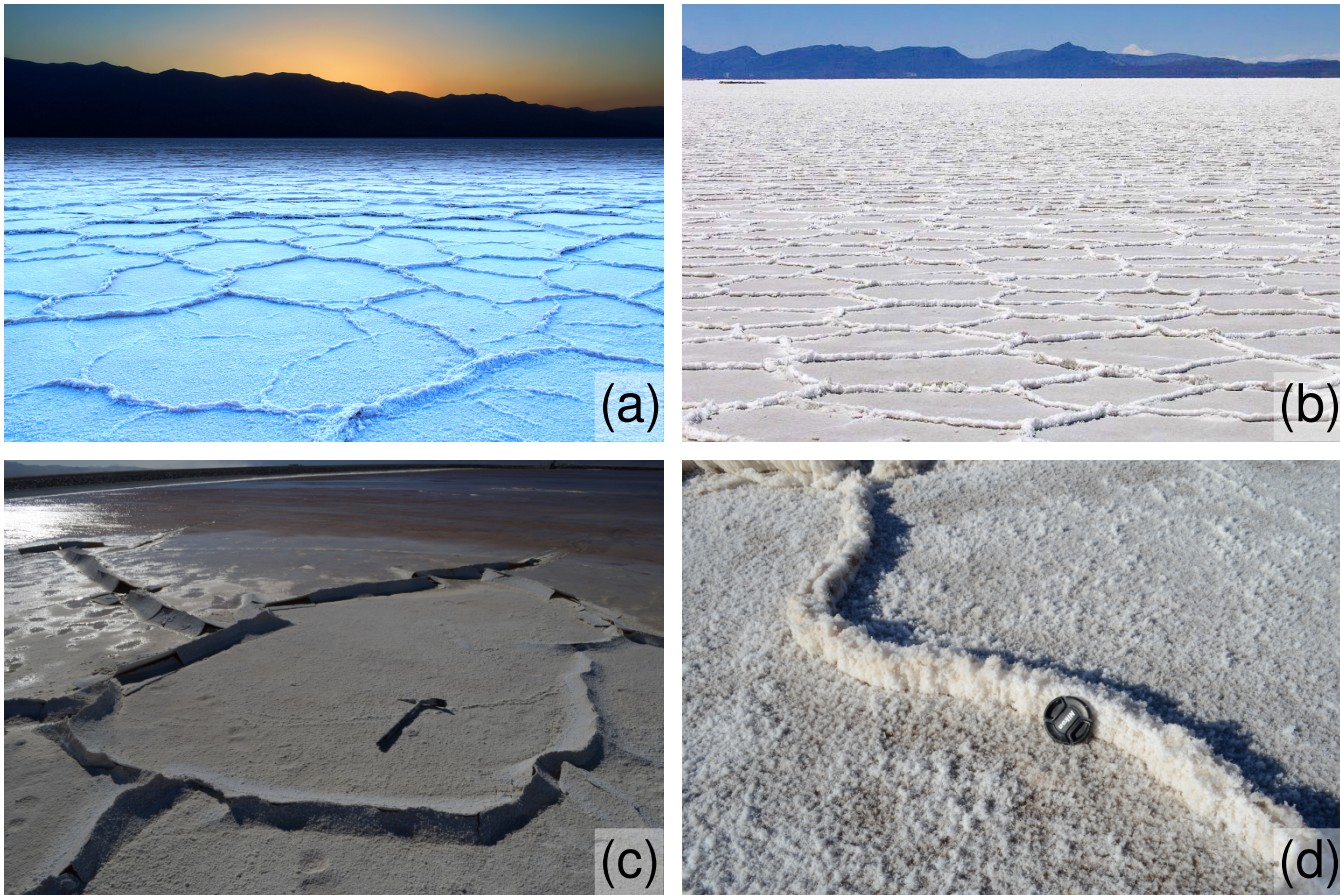

**Figure 1.** Polygonal ridge patterns in salt pans at (a) Badwater Basin, California (source: Photographersnature (2019)), (b) the Salar de Uyuni, Bolivia (source: Unel (2019)) and (c) Owens Lake, as well as (d) a close-up of a crust ridge at Badwater Basin.

### 2.1.1 Owens Lake

The Owens Lake basin is bounded by the Sierra Nevada fault zone to the west and the Inyo Mountain fault zone to the east (Hollet et al., 1991). The Owens Valley graben is deepest below Owens Lake: the valley fill reaches a depth of about $2.4\,$km above the bedrock (Hollet et al., 1991). The valley fill below the dry lake itself consists of moderately to well-sorted layers of sand with grain sizes that range between clay, fine to coarse sand and gravel (Hollet et al., 1991). The dry lake is framed by alluvial fan deposits. A more detailed description of the geology of the Owens Valley is given by Hollet et al.

(1991); Sharp and Glazner (1997) and Wilkerson et al. (2007).

All sampling locations at Owens Lake were situated in the area of the alluvial and lacrustine deposits (Hollet et al., 1991). The dry lake is divided into cells on which are implemented various dust control measures such as shallow flooding (Groeneveld and Barz, 2013), vegetation cover (Nicholas and Andy, 1997), gravel cover and encouraging salt crust growth in brine



cells (Groeneveld et al., 2010). We focused our sampling efforts on the brine cells in the north and south of the lake. Study
sites at Owens Lake are indicated in the map given in Fig. 2. For these sites we use labels referring to the surface management
cells of the dust control project there (LADWP, 2010). These labels either refer to managed cells or to unmanaged areas in the
direct vicinity of a managed cell. Labels start with TX-Y, where X is a number and Y is either a number or one of the letters
A, S and W. The first number refers to water taps (or turnoffs) along the main water pipeline that crosses the lake bed from
south to north and which is used to irrigate the managed area. Low tap numbers start in the south and the numbers generally
increase northwards. The second number refers to the $Y^{th}$ management cell connected to the $X^{th}$ turnoff. The letters A, W and
S refer to Addition, South and West, respectively; they also refer to different sub-regions branching from the same numbered
tap. Following the cell name is the letter P followed by a number which specifies an individual polygon sampled at that site.
For example, site label T27-A P3 refers to the third polygon sampled at the addition to the main cell at turnoff 27.

### 2.1.2 Badwater Basin

Badwater Basin is a geological sink and the lowest point on land in North America, about 86 m below sea level (Hunt et al.,
1966). Similar to Owens Lake it is subject to infrequent precipitation events and evaporation from the playa far outweighs
precipitation (Handford, 2003). Groundwater and runoff enter the basin from the surrounding mountains, carrying minerals
which accumulate in the basin floor (Hunt et al., 1966). The geology of the Badwater Basin and the surrounding Death Valley
is described in more detail elsewhere (Hunt et al., 1966; Sharp and Glazner, 1997) but, similar to Owens Lake, it also exhibits
a deep bed of unconsolidated valley fill, on which the salt crust rests.

We sampled polygons in an area about 500 m south of the main tourist pathway entering the salt flats from the east. This
area, chosen in consultation with park rangers, presented a convenient, typical and well-developed polygonal crust that was far
enough away from the tourist parking to minimise disturbances from other visitors. There, we sampled two polygons about
100 m inwards and parallel to the boundary of the salt flats. Additionally, we sampled three more polygons at distances of about
200 m, 300 m and 400 m inwards from the dry lake edge, respectively, to investigate any systematic effects of distance into the
salt pan. All sampling locations are depicted in Fig. 3.

### 2.2 Measurement protocols, instrumentation and sample analysis

### 2.2.1 Subsurface samples

We collected soil samples from below salt polygons using two different methodologies:

1) Digging a trench about 30 cm wide, 2 m long and 1 m deep and then collecting samples from one trench wall, as shown
in Fig. 4 (a).

2) Drawing cores with a Dutch gouge auger with a diameter of 50 mm and then collecting samples from the cores, as shown
in Fig. 4 (b). This method was used exclusively for wetter sites (water table within 30 cm of surface). The corer was
cleaned, rinsed with deionized water, and dried after each use.



For both sampling methodologies, we collected samples from directly below the crust to a depth of up to $1\,\mathrm{m}$. Samples were collected along a grid with a vertical resolution of approximately of $0.1$ to $0.15\,\mathrm{m}$ and a horizontal resolution of $0.15$ to $0.3\,\mathrm{m}$. Typically, sampling was done along a line passing through the middle of a polygon, included samples from directly under any bounding ridges and continued slightly into the two adjacent polygons. The samples had an average volume of approximately $10\,\mathrm{ml}$ and were taken using a metal spatula, which was cleaned with distilled water and dried before each use. The samples were a mixture of soil with a grain size of medium sand to clay, pore water and salt (both dissolved and precipitated). After collection, samples were immediately stored in air-tight containers, which were sealed with parafilm to prevent the loss of humidity between sample collection and measurement in the laboratory. Soil samples were then returned to the lab for further processing.

### 2.2.2 Grain size distributions

We measured the grain size distribution of the soil samples using a `Beckman Coulter LS 13 320` laser particle sizer (LPS). As preparation for this a soil sample would be thoroughly mixed with water, but without ultrasound treatment (i.e. we did not attempt to break up grain conglomerates). The resulting soil suspension was then pumped through the laser chamber of the LPS. The LPS measures the diffraction patterns generated as individual grains pass across the laser path, and these signals are converted into grain diameters $d_i$ based on Mie scattering theory (Hahn, 2009) with a real and imaginary component of $1.556$ and $0.1$, respectively; the underlying diffraction model we used was for quartz. By integrating many such measurements over time, a volume fraction $\varphi_i$ of grain diameters within a certain range – or bin – is calculated. Results are tabulated as the relative volume of particles within 93 distinct bins of particle diameters, which cover the range from $40\,\mathrm{nm}$ to $2000\,\mu\mathrm{m}$. The upper and lower cutoffs of each bin are given in Lasser and Goehring (2020b) along with this data.

Each grain size distribution measurement is an average of three independent measurements of the same sample. Even though there was no ultrasound treatment before measurement, there was no to minimal drift towards lower grain sizes due to dissolution of grain conglomerates over the three sequential measurements.

### 2.2.3 Salinity profiles

For each of the three trench sites (T32-1-L1 P2, T32-1-L1 P3 and T27-S P1) we compiled a cross-sectional salt concentration profile from samples taken from the trench wall, underneath a surface polygon. Samples collected by coring had insufficient sampling resolution to make similar cross-sectional profiles. We discuss the challenges encountered with measuring concentration profiles in more detail in Section 2.4. Samples were transported in sealed containers to a laboratory equipped with a high precision `Denver Instrument SI-234` balance with a precision of $\pm 0.1\,\mathrm{mg}$ as well as an oven to dry the samples. Gravimetric analysis of salt concentrations was conducted in the following steps:

– Extraction of the sample from its storage container into a crystallisation dish and measurement of the initial mass of the mixture of sand, salt and water.





- Drying of the sample in an oven at $80\,^{\circ}$C until all moisture had visibly vanished, or for at least 24 hours, followed by weighing to measure the amount of water that had evaporated from the sample as the difference from the sample mass before drying – i.e. to measure the initial water mass. Care was taken to let the samples cool down completely before weighing, because of the temperature sensitivity of the balance.

- Dilution of the sample with approximately $50\,$ml of deionized water followed by sedimentation of the solid sample components for roughly 24 hours and careful extraction of the supernatant liquid, which contains the dissolved salt, using a syringe. This step was repeated twice, and the extracted liquid was collected in a separate crystallisation dish.

- Separately drying the solid and liquid parts of the sample in an oven at $80\,^{\circ}$C until all moisture had visibly vanished or at least 24 hours had passed. For the liquid part, drying sometimes took considerably longer (e.g. a week). Finally, the
salt precipitated from the liquid phase and the now salt-free solid residue were individually weighed.

The comparison between the weight of the dry sample, composed of salt and sand, with the weight of the dry sample without the salt gives an *indirect* measure of the salt content in the sample whereas the weight of the salt crystallised in the dish gives a *direct* measure of the salt content. The difference between both weights gives an indication of the reliability of the analysis. If both weights were within the accuracy of the balance, the sample mass was conserved and the measurement was accepted.

**2.2.4    Salt crust and pore water samples**

The salt crusts observed at the dry lakes, and especially at Owens Lake, consisted of visually different patches of salt (see Fig. 5). This observation is consistent with the fact that silicates and carbonates have a significantly lower solubility than halite and thenardite and will tend to precipitate first as brine evaporates.

We collected samples from several sites, for subsequent chemical and mineral analysis; from each site we collected samples
from visibly different regions within the same site.

To collect saline pore water, we used a syringe to draw out the water which gathered in the coring holes. This worked well for the wetter field sites. For the dryer sites, where water did not readily gather in the holes, we used a perforated metal rod equipped with a filter and applied a negative pressure to suck water from the pores. In all cases, pore water was taken as close to the water table depth as possible.

**2.2.5    X-ray diffraction analysis of crust minerals**

To characterise the minerals present in the pore water and surface crust, samples were analysed using quantitative X-ray powder diffraction (XRD). The samples were prepared and analysed according to the following protocol:

- Similar to the procedure for measuring soil composition (Sec. 2.2.3), the total sample mass was first measured. Samples were then dried in an oven at $80\,^{\circ}$C for several days and the sample mass was measured again to determine the mass of
the evaporated water.



- Samples were mixed with 10 wt.% ZnO (zinc oxide) powder as a known baseline. This is necessary to quantify amorphous components in the sample.

- Samples were milled for 10 minutes in a `McCrown` micronising mill to create a fine powder.

- Samples were back-loaded into 27 mm sample holders to preserve a random crystal orientation in the powder.

- Samples were scanned in an X-ray diffractometer and diffraction patterns recorded.

- Minerals were identified using the `X'Pert HighScore` software (PANalytical).

- Mineral composition was quantified based on the Rietveld method using the software `AutoQuan` (Version 2.7.0.0).

For the XRD analysis we used a `Philips X'Pert MPD PW 3040` diffractometer, equipped with a PW 3050/10 goniometer, divergence slit of 0.5°, anti-scatter slit of 0.5°, receiving slit of 0.2 mm and a secondary graphite monochromator with a 20
160 mm mask, operating at 40 kV and 30 mA with Cu K$\alpha$ radiation. The range $2\theta = 4°$ to $69.5°$ was scanned with a step width of 0.02°. The counting time was 10 s and sample spinning was at 1 Hz. This produced measurements of the relative abundance of various salts at each sampled site.

### 2.2.6 Spectrometry analysis of crust minerals

To confirm the mineral quantifications measured by XRD, we used inductively coupled plasma optical emission spectrometry
(ICP-OES) to classify the ions in some of the pore water samples. Minerals from the pore water samples were first dried as described in Sec. 2.2.5. The residual salts were then re-dissolved in a known amount of water for measurement with ICP-OES.

All samples were analysed by an `Agilent 5100 VDV ICP-OES`. Ba, Ca, K, Na, and Sr were measured in radial view mode all other ions in axial view mode. Standardisation was done by using five matrix matched multi-element calibration solutions and a blank solution. All solutions contained $HNO_3$ at a concentration of 1.56 M. The mean of six blank measure-
170 ments was subtracted from each measurement. The detection limit was calculated as $3\sigma D_{BLANK}$, where $D_{BLANK}$ is the blank measurement.

### 2.2.7 Water density

To obtain a measurement of the water density, pore water samples were analysed in the laboratory using an `Anton Paar DMA 4500` vibrating-tube densitometer with a measurement accuracy of $5 \cdot 10^{-4}$ g/cm$^3$. For each sample, at least two measurements
were performed.

### 2.2.8 Temperature and relative humidity records

To characterise the environmental conditions at the field sites and how the surface crust introduces heterogeneity into these conditions, we embedded sensors into the crust and tracked their temperature and relative humidity over several days. Half of the sensors were placed inside hollow salt ridges (tepee structures) by carefully removing a small section of the salt crust



at the ridge, inserting the sensor into the hollow space inside the ridge and then putting the removed crust part back in place. The other sensors were embedded in the flat section of the salt crust in the middle of salt polygons. For these sensors, a small hole was dug into the crust and broken-up salt crystals were removed, the sensor was placed inside and then covered by the broken-up salt crystals. The measurements from within ridges can be compared to measurements of temperature and relative humidity recorded from sensors placed in the centres of polygons. For temperature and relative humidity measurements,

we used `HiTemp140` and `RHTemp1000IS` data loggers, which measured temperatures and relative humidity every $\Delta t = 60$ seconds with a precision of $\pm 0.01\,°\text{C}$ and $\pm 0.1\,\%$, respectively.

### 2.2.9   Surface scans using a terrestrial laser scanner (TLS)

To characterise the surface crust patterns, we recorded high resolution, three-dimensional point clouds of the surface relief using a `Leica P20` terrestrial laser scanner (TLS). From these, we extracted a characteristic pattern wavelength $\lambda$ and ridge

height $h$.

The scanner was positioned at a height of at least $2\,\text{m}$ above the crust (see Fig.4 (c)) and then recorded the surface relief in a circular sweep from a distance of about $1\,\text{m}$ from the scanner up to a distance of about $50\,\text{m}$. In principle the scanner can record the surface relief at even larger distances, but becomes increasingly prone to occlusion by surface features which lie in the path of the scanning beam. Additionally, the resolution of the scan decreases with distance from the scanner, as the number

of points measured along a circle with a given radius is fixed, even if the radius increases. Conversely, the resolution is highest in the direct vicinity of the scanner. Therefore we typically positioned the scanner not more then $10\,\text{m}$ away from the polygon we intended to sample, to record the pattern around the focal polygon with the highest possible accuracy.

At some sites it was possible to position the scanner on one of the gravel roads next to the site and therefore increase the vertical distance of the scanner to the crust to about $5\,\text{m}$. This was desirable, as it reduced occlusion at larger scanning distances.

At such sites we aimed to position the scanner such that the focal polygon was in the centre of the scanned area, which was a half-circle bounded by the gravel road. We aimed to scan each site we sampled before beginning the sampling procedure (i.e. before disturbing the naturally grown crust).

### 2.2.10   TLS Data processing

Scans acquired by the TLS were subject to several post-processing steps, following Nield et al. (2013) and described in detail

there. The main steps included: (1) extraction of a $10 \times 10\,\text{m}$ area of surface relief from the scan, including the sampled polygon at each site; (2) gridding of the data points into regular Cartesian coordinates using a nearest-neighbour algorithm implemented in Matlab; and (3) subtracting out the average background height of the surface. Due to the gridding into Cartesian coordinates, the best resolution of the surface relief in the processed data is $10\,\text{mm}$ in the horizontal and $0.3\,\text{mm}$ in the vertical direction.



### 2.2.11 Time lapse photography

To investigate the salt crust growth process, we recorded time lapse images of crusts at Owens Lake in spring 2018. We installed three cameras (`Ltl Acorn 5210A Trail Camera 940NM`) at different sites. All cameras were positioned in areas where shallow flooding was implemented as a dust control measure, and which were wet during our visit. We choose these sites since they promised to show active crust growth as temperatures increased and water evaporated during the course of the year.

We installed the cameras on tripods at a height of $1.5\,\mathrm{m}$ above the crust and secured them with rocks against strong winds. The battery-powered cameras recorded an image of a crust section of about $4\,\mathrm{m} \times 6\,\mathrm{m}$ in front of them every 30 minutes, from January $16^{th}$ to July $7^{th}$, 2018. We placed $0.1 \times 0.1\,\mathrm{m}$ white, grey and black tiles in the cameras' fields of view to allow us to calibrate the white balance and to act as a scale bar. Images were recorded with a resolution of 12 megapixels. Cameras were equipped with an infrared flash, which allowed them to record images during the night as well. Cameras were also equipped

with a temperature sensor, which recorded temperature with a precision of $\pm 1°$C. Since a camera's temperature sensor is embedded in its black casing, it overestimates temperatures during sunny days as the casing absorbs radiation and heats up. Date, time and temperature are encoded in the bottom of the recorded images.

Once installed, one camera completely failed to record any images. Of the remaining two cameras, about two-thirds of the images recorded by the cameras either failed to record completely or contained substantial digital artefacts. This may be due

to the rather harsh climate and other conditions at Owens Lake. To compare images with comparable lighting conditions, we handpicked images that recorded properly for every day shortly after sunrise and stitched them into a movie. Cameras stayed stationary during the whole period of recording and no re-alignment of images was necessary.

### 2.2.12 Pictures

During the two field trips, various pictures of the field sites, the salt crust and the sampling process were recorded using several

different cameras. From these images, a subset of images was selected to characterise each site.

### 2.3 Data provenance, structure and availability

An overview of all the available data is given in Table 2, indicating which type of data was collected and published from each site.

### 2.3.1 Grain size distributions

Data of grain size distributions are deposited at PANGAEA for 21 sites at Owens Lake and 5 sites at Badwater basin (Lasser and Goehring, 2020b). For each site, grain size distributions were measured for samples taken at different depths (for methods see Sec. 2.2.2). The overall soil composition is given as the dominant sample component following the Udden-Wentworth scale (Chesworth, 2008). The structure of the data sets is shown in Table 3.





| Site | Grain sizes | Salinity profile | Pore water density | Chemical analysis | Temp. data | RH data | TLS scan | Gridded TLS scan | Pictures | Video |
|---|---|---|---|---|---|---|---|---|---|---|
| Badwater Basin P1 | + | - | - | + | - | - | + | + | + | - |
| Badwater Basin P2 | + | - | - | + | - | - | + | + | + | - |
| Badwater Basin P3 | + | - | - | - | - | - | + | + | + | - |
| Badwater Basin P4 | + | - | - | - | - | - | + | + | + | - |
| Badwater Basin P5 | + | - | - | - | - | - | + | + | + | - |
| Owens Lake T2-4 P1 | + | - | - | + | - | - | + | + | + | - |
| Owens Lake T2-5 P1 | + | - | - | + | - | - | + | + | + | - |
| Owens Lake T2-5 P2 | + | - | - | + | - | - | + | + | + | - |
| Owens Lake T2-5 P3 | + | - | + | - | - | - | + | + | + | + |
| Owens Lake T8-W P1 | + | - | - | - | - | - | + | + | + | - |
| Owens Lake T10-3 P1 | + | - | + | + | - | - | + | + | + | - |
| Owens Lake T16 P1 | + | - | + | - | - | - | + | + | + | + |
| Owens Lake T25-3 P1 | + | - | - | - | - | - | + | + | + | - |
| Owens Lake T25-3 P2 | + | - | + | - | - | - | + | + | + | - |
| Owens Lake T27-A P1 | + | - | - | + | + | - | + | + | + | - |
| Owens Lake T27-A P2 | + | - | - | + | - | - | + | - | + | - |
| Owens Lake T27-A P3 | + | - | + | - | - | - | + | + | + | - |
| Owens Lake T27-S P1 | + | + | - | - | - | - | + | + | + | - |
| Owens Lake T29-3 P1 | + | - | - | + | + | + | + | + | + | - |
| Owens Lake T29-3 P2 | + | - | - | + | - | - | + | - | + | - |
| Owens Lake T32-1-L1 P1 | + | - | - | + | - | - | + | + | + | - |
| Owens Lake T32-1-L1 P2 | + | + | + | - | - | - | + | - | + | - |
| Owens Lake T32-1-L1 P3 | + | + | + | - | - | - | + | + | + | - |
| Owens Lake T36-3 P1 | + | - | - | + | + | + | + | + | + | - |
| Owens Lake T36-3 P2 | + | - | - | + | - | - | + | + | + | - |
| Owens Lake T36-3 P3 | + | - | - | + | - | - | + | + | + | - |

**Table 2.** Availability of data sets for each of the 5 field sites at Badwater Basin and the 21 field sites at Owens Lake. Columns indicate data sets for grain size distributions (Lasser and Goehring, 2020b), salt concentration profiles and pore water density measurements (Lasser and Goehring, 2020a), chemical analysis of the pore water and salt crust components (Lasser and Karius, 2020), temperature and relative humidity (RH) time series (Nield et al., 2020a), raw and gridded TLS surface scan data (Nield et al., 2020b) as well as pictures and time-lapse videos of the field sites (Lasser et al., 2020). All data sets are available at PANGAEA.

### 2.3.2 Cross-sectional salinity profiles

Data of subsurface salinity distributions are available at PANGAEA for the three trench sites at Owens Lake (Lasser and Goehring, 2020a). The lateral position of the sample refers to the distance from the first sample taken along a line that bisects the main sampled polygon, and extends slightly into the two adjacent polygons. The positions of the salt ridges at the surface for the sites are given in Table 4.





| # | Name | Short name | Unit | Note |
|---|------|-----------|------|------|
| 1 | DEPTH, sediment/rock | Depth | m | Geocode |
| 2 | Percentile 10 | Perc10 | $\mu$m | |
| 3 | Median, grain size | D50 | $\mu$m | |
| 4 | Percentile 90 | Perc90 | $\mu$m | |
| 5 | Soil composition | Soil comp | | |
| 6 | Soil water content | Soil water | % | wt% |
| 7 | Diameter | ø | $\mu$m | lower channel diameter |
| 8 | Difference | Diff | % | volume |
| 9 | Difference | Diff | % | in volume -2SD |
| 10 | Difference | Diff | % | in volume +2SD |
| 11 | Diameter | ø | $\mu$m | middle channel diameter |
| 12 | Diameter | ø | $\mu$m | upper channel diameter |
| 13 | Diameter | Diam | phi | middle channel diameter |
| 14 | Difference | Diff | % | volume cumulated |

**Table 3.** Structure of the grain size analysis data sets at PANGEA (Lasser and Goehring, 2020b). Column names identify the depth at which each sample was collected, a lower ($10^{th}$ percentile), median and upper ($90^{th}$ percentile) representative value of the grain size distribution, a general soil classification based on the most ubiquitous grain size range (following Chesworth (2008)) and the weight % of water in the soil sample. Column names also identify the lower channel diameter of the laser particle analyser (in $\mu$m), the differential volume recorded in the respective channel and the differential volume minus (plus) two standard deviations. Additionally, the middle and upper channel diameter (in $\mu$m), the middle channel diameter in the Krumbein $\phi$ scale (Krumbein, 1934) and the cumulated differential volume are reported.





| Site | First ridge [cm] | Second ridge [cm] |
|------|------------------|-------------------|
| T27-S | 30 | 210 |
| T32-1-L1 P2 | 45 | 195 |
| T32-1-L1 P3 | 30 | 210 |

**Table 4.** Positions of the polygon ridges at the trench sites for which subsurface salinity profiles were collected, relative to the start of the sampling transect.

| # | Name | Short name | Unit | note |
|---|------|------------|------|------|
| 1 | Sample code/label | Sample label | | |
| 2 | Z Axis | Z | cm | |
| 3 | X Axis | X | cm | |
| 4 | Water content, wet mass | Water wm | g | |
| 5 | Sand, mass netto | Sand | g | |
| 6 | Salt content | Salt | g | direct |
| 7 | Salt content | Salt | g | indirect |

**Table 5.** Structure of the subsurface salinity profile data sets at PANGEA (Lasser and Goehring, 2020a). The data contain sample labels, the vertical (Z) and horizontal (X) position of each sample along the survey and the masses of water, sand and salt (measured by two methods) in those samples. See Section 2.2.3 for details of analysis methods.

The data was collected during a field campaign in January 2018. Gaps in the data are due to either contamination of samples
with surface salt or loss of samples during the destructive analysis process. The structure of the data sets is shown in Table 5.

### 2.3.3 X-ray diffraction data

Data of the composition of the salt crust analysed via quantitative X-ray diffraction are available at PANGAEA for 10 samples
from 2 sites at Owens Lake (Lasser and Karius, 2020). Samples were collected in 2016. The structure of the data sets is shown
in Table 6.

**2.3.4 Spectrometry data**

Data of the composition of the pore water analysed via ICP-OES are available at PANGAEA for 10 sites at Owens Lake and
two sites at Badwater Basin (Lasser and Karius, 2020). Samples were collected during 2016. The ICP-OES analysed for the
ions of 29 distinct elements in solution. The structure of data sets is shown in Table 7.



| # | Name | Short name | Unit | Note |
|---|------|-----------|------|------|
| 1 | Event label | Event | | |
| 2 | Site | Site | | |
| 3 | Sample ID | Sample ID | | |
| 4 | Percentage | Perc | % | Trona, $Na_3(HCO_3)(CO_3)\cdot 2H_2O$ |
| 5 | Percentage | Perc | % | Halite, $NaCl$ |
| 6 | Percentage | Perc | % | Burkeite, $Na_6(CO_3)(SO_4)_2$ |
| 7 | Percentage | Perc | % | Thenardit, $Na_2SO_4$ |
| 8 | Percentage | Perc | % | Calcite, $CaCO_3$ |
| 9 | Percentage | Perc | % | Albite, $NaAlSi_3O_8$ |
| 10 | Percentage | Perc | % | Nahcolite, $NaHCO_3$ |
| 11 | Percentage | Perc | % | Janhaugite, $(Na,Ca)_3(Mn,Fe)_3(Ti,Zr,Nb)_2Si_4O_{16}(OH,F)_2$ |
| 12 | Percentage | Perc | % | Quartz, $SiO_2$ |
| 13 | Percentage | Perc | % | Orthoclase, $KAlSi_3O_8$ |
| 14 | Percentage | Perc | % | Siderophyllite, $KFe_2Al(Al_2Si_2)O_{10}(F,OH)_2$ |
| 15 | Percentage | Perc | % | Huntite, $Mg_3Ca(CO_3)_4$ |
| 16 | Percentage | Perc | % | $CaHPO_4$ |
| 17 | Percentage | Perc | % | Senarmontite, $Sb_2O_3$ |
| 18 | Percentage | Perc | % | $(Mg,Fe,Al)_6(Al,Si)_4O_{10}(OH)_8$ |
| 19 | Percentage | Perc | % | Portlandite, $Ca(OH)_2$ |

**Table 6.** Structure of the salt crust salt species characterisation via quantitative X-ray diffraction data set at PANGEA (Lasser and Karius, 2020). The data consist of site and sample IDs and the weight percentages of various salts within each sample.

### 2.3.5 Water density data

Data of pore water density are available at PANGAEA for 7 sites at Owens Lake (Lasser and Goehring, 2020a). Individual measurements of densities are reported, along with averages and standard deviations for all measurements on individual samples. The structure of data sets is shown in Table 8.

### 2.3.6 Temperature and relative humidity recordings

Data of the temperature and relative humidity recordings are available at PANGAEA for 3 sites at Owens Lake (Nield et al., 2020a). The data was collected from November 25[th] to December 2[nd] 2016. The structure of data sets is shown in Table 9. Note that not all tables contain a humidity column, since some sensors were only able to record temperature.



| # | Name | Short name | Unit |
|---|------|-----------|------|
| 1 | Event label | Event | |
| 2 | Site | Site | |
| 3 | Aluminium | Al | μg/l |
| 4 | Arsenic | As | μg/l |
| 5 | Barium | Ba | μg/l |
| 6 | Calcium | Ca | μg/l |
| 7 | Cadmium | Cd | μg/l |
| 8 | Cerium | Ce | μg/l |
| 9 | Cobalt | Co | μg/l |
| 10 | Chromium | Cr | μg/l |
| 11 | Copper | Cu | μg/l |
| 12 | Iron | Fe | μg/l |
| 13 | Gadolinium | Gd | μg/l |
| 14 | Potassium | K | μg/l |
| 15 | Lanthanum | La | μg/l |
| 16 | Lithium | Li | μg/l |
| 17 | Magnesium | Mg | μg/l |
| 18 | Manganese | Mn | μg/l |
| 19 | Molybdenum | Mo | μg/l |
| 20 | Sodium | Na | μg/l |
| 21 | Niobium | Nb | μg/l |
| 22 | Nickel | Ni | μg/l |
| 23 | Phosphorus | P | μg/l |
| 24 | Lead | Pb | μg/l |
| 25 | Platinum | Pt | μg/l |
| 26 | Sulphur | S | μg/l |
| 27 | Scandium | Sc | μg/l |
| 28 | Strontium | Sr | μg/l |
| 29 | Titanium | Ti | μg/l |
| 30 | Vanadium | V | μg/l |
| 31 | Zinc | Zn | μg/l |

**Table 7.** Structure of the pore water ion characterisation via ICP-OES data set at PANGEA (Lasser and Karius, 2020). The data gives sample locations and the concentration of various ions dissolved in the water samples.



| # | Name | Short name | Unit | Note |
|---|---|---|---|---|
| 1 | Event label | Event | | |
| 2 | Latitude of event | Latitude | | |
| 3 | Longitude of event | Longitude | | |
| 4 | Site | Site | | |
| 5 | Sample ID | Sample ID | | |
| 6 | DEPTH, sediment/rock | Depth | m | |
| 7 | DISTANCE | Distance | cm | |
| 8 | Density, pore water | Dens pw | g/cm$^3$ | measure 1 |
| 9 | Density, pore water | Dens pw | g/cm$^3$ | measure 2 |
| 10 | Density, pore water | Dens pw | g/cm$^3$ | measure 3 |
| 11 | Density, pore water | Dens pw | g/cm$^3$ | measure 4 |
| 12 | Density, pore water | Dens pw | g/cm$^3$ | average |
| 13 | Density, standard error | Density std e | $\pm$ | |

**Table 8.** Structure of the pore water density data sets at PANGEA (Lasser and Goehring, 2020a). The data include site and sample labels, the depth of each water sample (typically at the water table height) and its horizontal location along the transect, as well as all individual density measurements and the averages and standard deviations for each sample.

| # | Name | Short name | Unit |
|---|---|---|---|
| 1 | DATE/TIME | Date/Time | |
| 2 | Temperature, air | TTT | °C |
| 3 | Humidity, relative | RH | % |

**Table 9.** Structure of temperature and relative humidity data sets at PANGEA (Nield et al., 2020a). Data are taken at the ridges and polygon centres of three sites, every 2 minutes from November $25^{th}$ to December $2^{nd}$, 2016.





| # | Name | Short name | Unit | Note |
|---|------|-----------|------|------|
| 1 | Event label | Event | | |
| 2 | Latitude of event | Latitude | | |
| 3 | Longitude of event | Longitude | | |
| 4 | File content | Content | | |
| 5 | File format | File format | | gridded |
| 6 | File size | File size | kByte | gridded |
| 7 | Uniform resource locator/link to file | URL file | | gridded |
| 8 | File format | File format | | raw |
| 9 | File size | File size | kByte | raw |
| 10 | Uniform resource locator/link to file | URL file | | raw |

**Table 10.** Structure of the surface scan data set collection of raw point clouds and gridded subsets at PANGEA (Nield et al., 2020b). For each scan there is also an associated file containing a full 3D point cloud of the surface scan, with links embedded in the data table.

### 2.3.7 Surface scans

Raw 3D point clouds recorded with a TLS are available at PANGAEA for all sites at Owens Lake and Badwater Basin. The raw point cloud data sets each contain a list of points containing coordinates in the format of ($x$-position, $y$-position, elevation). These data are georeferenced, and give easting ($x$-position) and northing ($y$-position) within the U.S. National Grid (USNG). For Owens Lake, these positions are relative to grid zone and square 11S MA, whereas locations at Badwater Basin are relative to 11S NA. Elevations are referenced to the WGS 84 geoid.

Gridded subsets of the point clouds are available for all sites at Badwater Basin and for 18 sites at Owens Lake (Nield et al., 2020b), as listed in Table 2. These data sets are matrices where each entry represents an elevation. Points are regularly spaced with a resolution of $\Delta X = \Delta Y = 0.01\,\mathrm{m}$.

The structure of the tables further detailing both types of TLS data is shown in Table 10.

### 2.3.8 Pictures

Pictures are available at PANGAEA for all 21 sites at Owens Lake and 5 sites at Badwater Basin (Lasser et al., 2020). The data set also contains two time-lapse videos of sites T16 P1 and T2-5 P3 at Owens Lake. The set of images for each field site contains:

- three images of the field site and its general surroundings,
- up to three images of single polygons with a scale bar at the field site,



| Item | Note | Size [m] |
|------|------|----------|
| lens cover | shortest diameter | 0.052 |
| red rock hammer | pick length | 0.18 |
| red rock hammer | full length | 0.31 |
| blue pick axe | pick length | 0.33 |
| blue pick axe | full length | 0.55 |
| folding rule | folded | 0.24 |

**Table 11.** Sizes of various items included for scale in images from the data-set given by Lasser et al. (2020).

| # | Name | Short name |
|---|------|-----------|
| 1 | Event label | Event |
| 2 | Latitude of event | Latitude |
| 3 | Longitude of event | Longitude |
| 4 | File content | Content |
| 5 | File name | File name |
| 6 | File format | File format |
| 7 | File size | File size |
| 8 | Uniform resource locator/link to image | URL image |
| 9 | Uniform resource locator/link to movie | URL movie |

**Table 12.** Structure of the image and video collection of field sites at PANGEA (Lasser et al., 2020). The data table contains links to the individual images and time-lapse movies.

- one image of the sampled polygon with an indication of the sampling positions (either small holes in the crust or orange markers),

- one image of the sampled core, and

- up to 8 images of salt crust features.

The sizes of various items included for scale in the images are given in Table 11. The structure of the data set collection is shown in Table 12.

## 2.4 Results

Here we present sample data from each of the different data sets, in order to illustrate the kind and quality of information contained in them.



### 2.4.1 Grain size distributions

Exemplary grain size distributions are shown in Fig. 6 for Owens Lake (Fig. 6(a)) and Badwater Basin (Fig. 6(b)). These show how the distribution of particle sizes changes with depth at a representative site from each lake. Depending on the site, grain size distributions often show a pronounced layering of the soil, featuring widely varying multi-modal grain size distributions. This is indicative of a phased sand deposition process (Earle, 2015, p. 361) and consistent with the history of flooding following heavy rainfall at both Owens Lake and Badwater Basin.

### 2.4.2 Salinity profiles

As mentioned in Section 2.2.3, we used two different methodologies during sample collection: at drier sites (with a water table at approximately $0.7\,\mathrm{m}$) we dug trenches, whereas at wetter sites with a water table nearer to the salt crust (typically at a depth of $0\,\mathrm{m}$ to $0.3\,\mathrm{m}$) we extracted soil cores using a Dutch gouge auger. The sampling from trenches yielded much more reliable results than the sampling from cores. Consequently, only samples from trench sites were used to compile salinity profiles. One example of a salinity profile compiled from samples extracted from a trench is given in Fig. 7

Analysis of many of the wet sites showed that the coring method introduced significant noise into the salt content measurements. We identified two main sources of this noise. Firstly, salt crystals (often from the crust) could be pushed into the soil by the corer and subsequently positioned away from their original depths. Even small displaced salt crystals are enough to considerably disturb the measurements of salt content. Secondly, water from close to the surface, and presumably with a high salt concentration, was often seen to be running down the corer after we pulled it out of the ground. We tried to prevent contamination by sampling with the core laid out horizontally, and by removing an outer layer of approximately $5\,\mathrm{mm}$ from the surface of the core prior to sample collection. Nevertheless, especially for high permeability soils, the core was likely contaminated by brine to some degree.

Additionally, during the 2016 field campaign, we sampled the soil with a lower horizontal resolution. As a consequence, results from sites where we used the corer and where we collected samples with a horizontal resolution lower than $\Delta X = 0.2\,\mathrm{m}$ were not used for the analysis of concentration gradients.

We also performed a reproducibility trial of the salinity profiles by collecting replicate samples at one site were samples were collected via the corer. The maximum deviation in salinity between the samples collected right next to each other was 5.7 wt.%. Excluding the shallowest two samples rows of samples (where contamination from surface crust pieces is likely), the maximum deviation is about 2.5 wt.% and the mean deviation is about 1 wt.%.

### 2.4.3 Chemical analysis

The most abundant salts found in the different crust samples collected from two sites at Owens Lake are listed in Table 13. The samples, taken from visually distinct patches of salt that were nonetheless near to each other, show different compositions. This presumably reflects how the various salts in solution will start to crystallise at different times in the brine evaporation process.





| Site | Sample | Component 1 | % | Component 2 | % |
|------|--------|-------------|---|-------------|---|
| T2-5 | 1 | Calcite, $CaCO_3$ | 35 | Albite, $NaAlSi_3O_8$ | 35 |
| T2-5 | 2 | Burkeite, $Na_6(CO_3)(SO_4)_2$ | 52 | Nahcolite, $NaHCO_3$ | 23 |
| T2-5 | 3 | Thenardite, $Na_2SO_4$ | 46 | Trona, $Na_3(HCO_3)(CO_3) \cdot 2H_2O$ | 29 |
| T2-5 | 4 | Trona, $Na_3(HCO_3)(CO_3) \cdot 2H_2O$ | 73 | Burkeite, $Na_6(CO_3)(SO_4)_2$ | 15 |
| T2-5 | 5 | Trona, $Na_3(HCO_3)(CO_3) \cdot 2H_2O$ | 76 | Burkeite, $Na_6(CO_3)(SO_4)_2$ | 15 |
| T2-5 | 6 | Trona, $Na_3(HCO_3)(CO_3) \cdot 2H_2O$ | 47 | Halite, $NaCl$ | 25 |
| T10-3 | 1 | Halite, $NaCl$ | 33 | Trona, $Na_3(HCO_3)(CO_3) \cdot 2H_2O$ | 31 |
| T10-3 | 2 | Halite, $NaCl$ | 100 | | |
| T10-3 | 3 | Halite, $NaCl$ | 100 | | |
| T10-3 | 4 | Trona, $Na_3(HCO_3)(CO_3) \cdot 2H_2O$ | 40 | Thenardite, $Na_2SO_4$ | 30 |

**Table 13.** List of the most and second-most abundant salt species in crust samples taken at sites T2-5 and T10-3. The full data set is available at Lasser and Karius (2020).

At Owens Lake the analysis of pore water ions via ICP-OES is dominated by sodium, sulphur and potassium (in descending level of significance), but also shows notably high levels of arsenic, of up to $150\,\mu$g/l. This is consistent with other reports of
arsenic found in the salt crust (Ryu et al., 2002; Gill et al., 2002).

### 2.4.4 Water density

Measurements of the density of water samples collected from different depths allows for a reliable quantification of the background salinity at Owens Lake. A comparison between the background salinity and surface salinity then allows for an estimation of the buoyancy forces that the more saline, and therefore heavier, water at the surface is subjected to. Water samples
collected from at the surface (or just under the crust) typically show density values of approximately $1.21\,$g/ml, whereas at a depth of about $0.9\,$m the salt water density is approximately $1.05\,$g/ml. This is consistent with water density measurements performed by Tyler et al. (1997). Both Owens Lake and Badwater Basin show a salinity distribution that should allow for the convective overturning of their pore water (Wooding et al., 1997; Lasser et al., 2019).

### 2.4.5 Surface scans

Scans of salt pan surfaces, using a high resolution terrestrial laser scanner, allow for a quantification of the pattern dimensions. Gridded subsets consisting of the three-dimensional point clouds of various $\sim 10\,$m $\times$ $10\,$m areas are shown in Fig. 8. The average pattern wavelengths, $\lambda$, and average ridge heights, $h$, for each site were calculated from the gridded scans and are given in Table 14. Uncertainties for $\lambda$ and $h$ are given as the standard deviations of the measured pattern wavelength and height in the gridded subset, respectively. Values for the pattern wavelength consistently lie in the range of $0.5\,$m to $3\,$m. Furthermore,
the pattern wavelength is weakly but positively correlated with polygon height ($R^2 = 0.31$, $p = 0.004$). The values for the wavelength can also be compared to models of subsurface convective motion (see for example Lasser et al. (2019)).





| Site | $\lambda$ [m] | $h$ [$10^{-2}$m] |
|---|---|---|
| Badwater Basin P1 | $1.42 \pm 0.58$ | $7.7 \pm 2.8$ |
| Badwater Basin P3 | $1.27 \pm 0.55$ | $7.1 \pm 2.6$ |
| Badwater Basin P4 | $0.58 \pm 0.32$ | $2.8 \pm 1.3$ |
| Badwater Basin P5 | $0.55 \pm 0.28$ | $3.4 \pm 1.4$ |
| T10-3 P1 | $1.79 \pm 0.86$ | $7.4 \pm 3.1$ |
| T16 P1 | $1.39 \pm 0.51$ | $7.5 \pm 2.8$ |
| T2-4 P1 | $1.13 \pm 0.54$ | $2.5 \pm 1.0$ |
| T2-5 P1 | $1.04 \pm 0.41$ | $4.5 \pm 1.6$ |
| T2-5 P2 | $0.94 \pm 0.50$ | $2.6 \pm 1.5$ |
| T2-5 P3 | $1.62 \pm 0.65$ | $4.4 \pm 1.6$ |
| T25-3 P1 | $2.25 \pm 0.89$ | $15.3 \pm 5.1$ |
| T25-3 P2 | $1.18 \pm 0.56$ | $6.2 \pm 2.2$ |
| T27-A P1 | $1.70 \pm 0.65$ | $5.0 \pm 1.8$ |
| T27-A P2 | $2.72 \pm 0.98$ | $7.6 \pm 2.5$ |
| T27-A P3 | $1.44 \pm 0.55$ | $7.3 \pm 2.4$ |
| T27S P1 | $1.51 \pm 0.64$ | $6.5 \pm 2.4$ |
| T29-3 P1 | $3.02 \pm 1.40$ | $7.3 \pm 3.1$ |
| T29-3 P2 | $2.80 \pm 1.34$ | $6.7 \pm 3.0$ |
| T32-1-L1 P1 | $1.56 \pm 0.66$ | $13.8 \pm 4.8$ |
| T32-1-L1 P2 | $2.65 \pm 0.98$ | $10.8 \pm 3.6$ |
| T32-1-L1 P3 | $2.43 \pm 0.92$ | $7.8 \pm 2.8$ |
| T36-3 P1 | $1.17 \pm 0.91$ | $2.2 \pm 2.2$ |
| T36-3 P2 | $2.27 \pm 1.03$ | $7.2 \pm 3.4$ |
| T36-3 P3 | $1.43 \pm 0.62$ | $4.9 \pm 1.9$ |
| T8-W P1 | $0.87 \pm 0.41$ | $3.9 \pm 1.5$ |

**Table 14.** Average pattern wavelengths ($\lambda$) and ridge heights ($h$) calculated from surface scans of salt pans showing polygonal shapes. Ranges indicate the standard deviation of each set of measurements.

### 2.4.6 Videos

The two videos that were successfully compiled from time lapse photography at sites Owens Lake T16 P1 and Owens Lake T2-5 P3 show the growth of the salt crust from a flooded configuration. Growth starts as soon as the water table sinks below the crust surface, in late March and early April. Ridges seem to preferentially grow in locations were ridges were present before the flooding. From the videos, salt ridge growth of about 30-50 mm over the course of 20 days can be inferred, i.e. a rate of about 2 mm / day (see video from site Owens Lake T16 P1, lower right corner), which is similar to the growth rates observed for salt pans in Botswana (Nield et al., 2015).





## 3 Summary

Six data sets were presented which characterise the surface, subsurface and environmental conditions of two dry salt lakes – Owens Lake and Badwater Basin – in central California. The data sets include grain size distribution measurements of sand samples taken at these locations (Lasser and Goehring, 2020b), subsurface cross-sectional salt concentration profiles and pore water density measurements (Lasser and Goehring, 2020a), a chemical characterisation of the various salts present in the salt crust and pore water (Lasser and Karius, 2020), temperature and relative humidity measurements from within salt ridges and

polygon centres (Nield et al., 2020a), high resolution surface scans measured using a terrestrial laser scanner (Nield et al., 2020b) as well as images characterising the field sites and time-lapse videos capturing the growth of salt polygons (Lasser et al., 2020).

Grain size distributions, surface scans and images cover all 26 sites at Badwater Basin and Owens Lake that were visited and allow for an in-depth characterisation of surface and sub-surface conditions at these salt pans. Temperature and relative

humidity recordings are only available for three sites at Owens Lake but allow for an estimation of the impact of the presence of salt ridges on temperature and humidity and therefore evaporation of water from the crust. Videos were compiled at two sites at Owens Lake and are direct evidence of salt ridges growing on a very short time scale. The analysis of salt species only covers 2 sites at Badwater Basin and 6 at Owens lake. Nevertheless it is to be expected that other sites in the area have a similar mineral composition, since they are connected to the same ground water reservoirs.

For future research, the environmental conditions at the salt polygons and inside the salt ridges could be better described, since they are closely linked to evaporative processes in these landscapes. Evaporation is both important for the water and energy balance of salt pans and as a driver of potential dynamical processes below the crust. Measurements of temperature and relative humidity could also be accompanied by direct measurements of the evaporation rate.

Furthermore, properties of the salt crust itself, such as crust thickness, would likely be of interest to investigate. This is

365 important since theories about the origin of salt polygons make statements about the preferential deposition of salt in certain parts of polygons. Data about the crust thickness at salt ridges as compared to the centre of polygons could help confirm these theories.

## 4 Data availability

The data set has been split up into five subsets available under the DOIs listed below. The different data types described in

this publication are very different regarding content and size, therefore we decided to split the data set into different subsets to facilitate dissemination and communication.

Grain size distributions are available at https://doi.pangaea.de/10.1594/PANGAEA.910996 (Lasser and Goehring, 2020b). Salt concentration profiles and pore water density measurements are available at https://doi.pangaea.de/10.1594/PANGAEA. 911059 (Lasser and Goehring, 2020a). Results of the chemical characterization of salts present at the sites are available at

375 https://doi.pangaea.de/10.1594/PANGAEA.911239 (Lasser and Karius, 2020). Temperature and humidity recordings are available at https://doi.pangaea.de/10.1594/PANGAEA.911139 (Nield et al., 2020a). Raw and post-processed surface scan data are





available at https://doi.pangaea.de/10.1594/PANGAEA.911233 (Nield et al., 2020b). Images and videos of the field sites are available at https://doi.pangaea.de/10.1594/PANGAEA.911054 (Lasser et al., 2020)

*Author contributions.* JL was responsible for the conduction of the two field campaigns, laser particle size measurements, laboratory analysis
of all samples and preparation of the manuscript. LG was responsible for the conception of the project, performed the pore water density measurements, participated in both field campaigns and contributed to the writing of the manuscript. JN participated in both field campaigns and was responsible for the conduction and post-processing of TLS measurements. All authors contributed to proofreading of the manuscript.

*Competing interests.* The authors declare no competing interests.

*Acknowledgements.* We thank Grace Holder (Great Basin Unified Air Pollution Control District) for support at Owens Lake and the U.S.
National Park Service for access to Death Valley (Permit DEVA-2016-SCI-0034). TLS processing used the Iridis Southampton Computing Facility. We thank Volker Karius of the Geowissenschaftliches Zentrum, Georg-August-University Göttingen for help with the conduction of the quantitative XRD and ICP-OES measurements. We thank the Stifterverband, Wikimedia Germany and the Volkswagen foundation for partly funding JL during the work on this publication.



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



Owens River

● concentration profile

● samples

**N**

**W** ⊕ **E**

**S**

managed
area

**T32-1-L1**

**T29-3**

Keeler

**T36-3**

**T27-A**

Owens Lake
(dry)

**T27-S**

**T27-3**

main gravel road

unmanaged
area

managed
area

California

San
Francisco

Los
Angeles

**T16**

**T10-3**

**T8-W**

managed
area

**T2-5**

**T2-4**

2 km





**Figure 3.** Map of Badwater Basin in the Death Valley, central California, USA. Sampling sites are indicated as red dots.
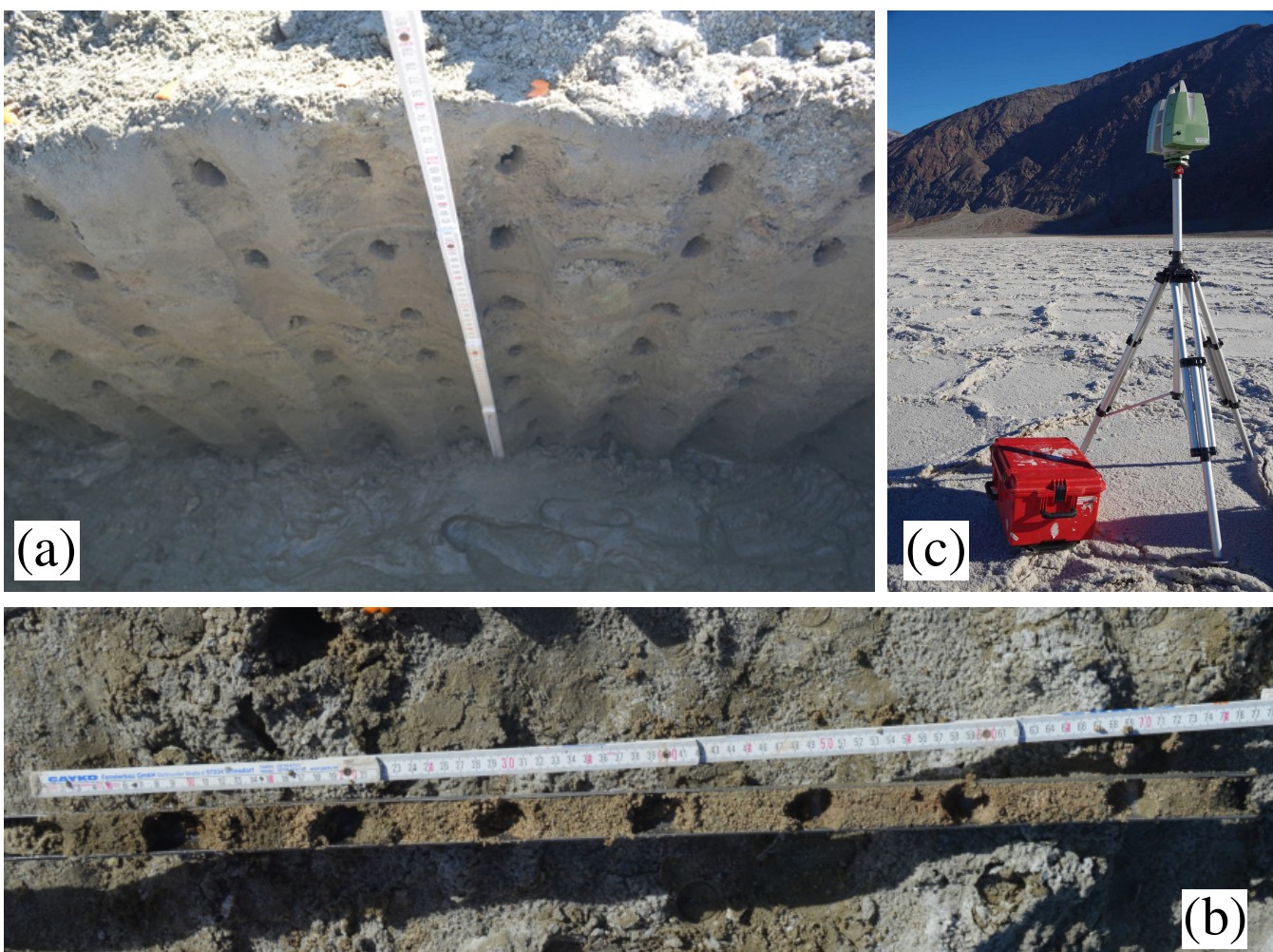

**Figure 4.** Field methods. (a) Representative trench and sampling positions (holes) at a site at Owens Lake. (b) Dutch gouge auger and sampling positions (holes) at a site at Owens Lake (crust-soil interface is positioned at 0 cm). (c) Terrestrial Laser Scanner (TLS) setup at Badwater Basin.

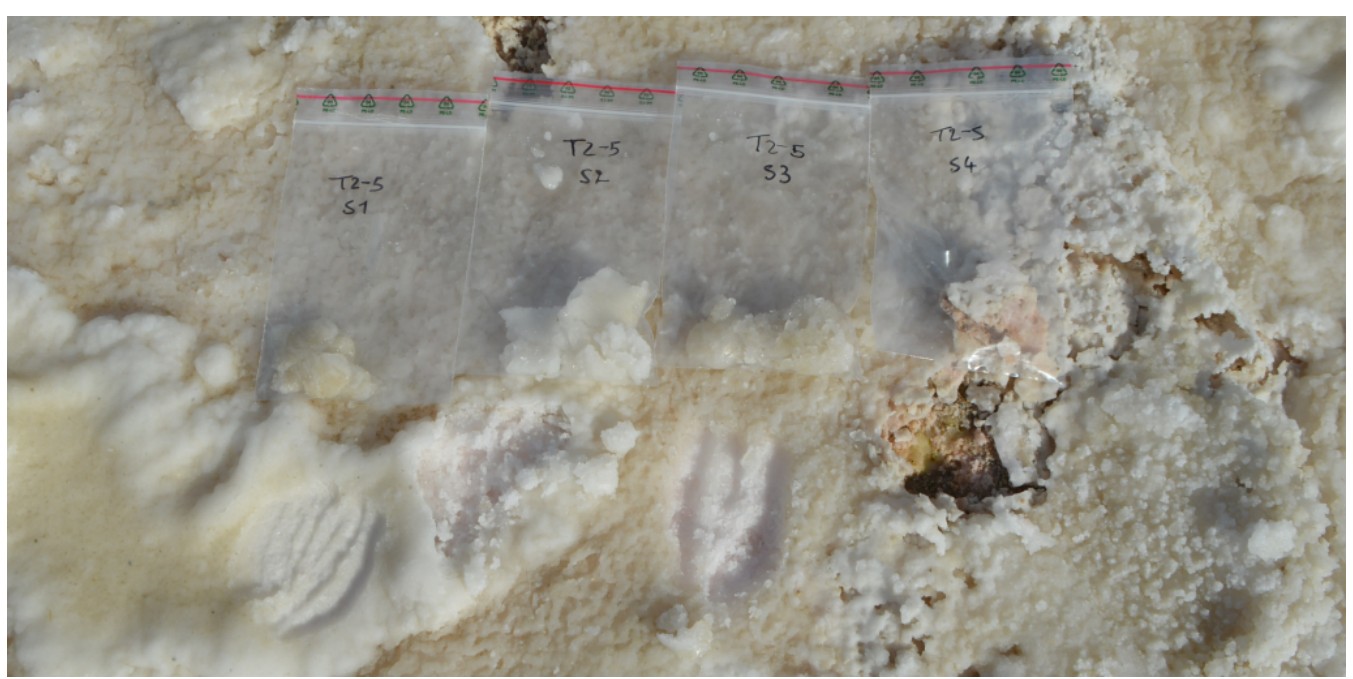

**Figure 5.** Examples of salt samples collected from the salt crust at site T2-5 P3 at Owens Lake, California.

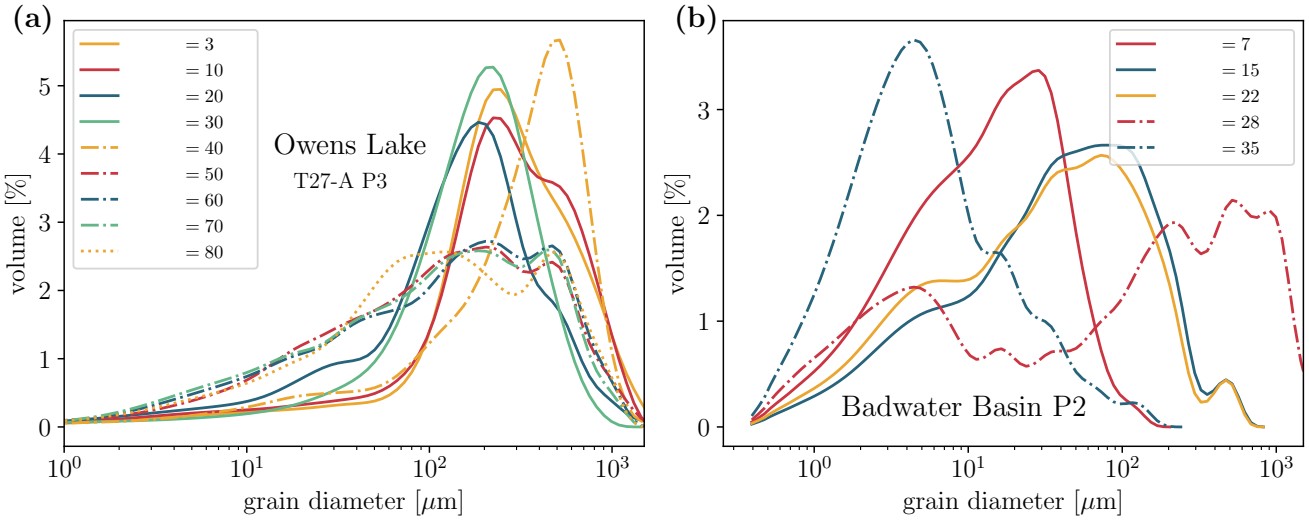

**Figure 6.** Exemplary grain size (diameter) profiles for **(a)** site T27-A P3 at Owens Lake and **(b)** site Badwater Basin P2.



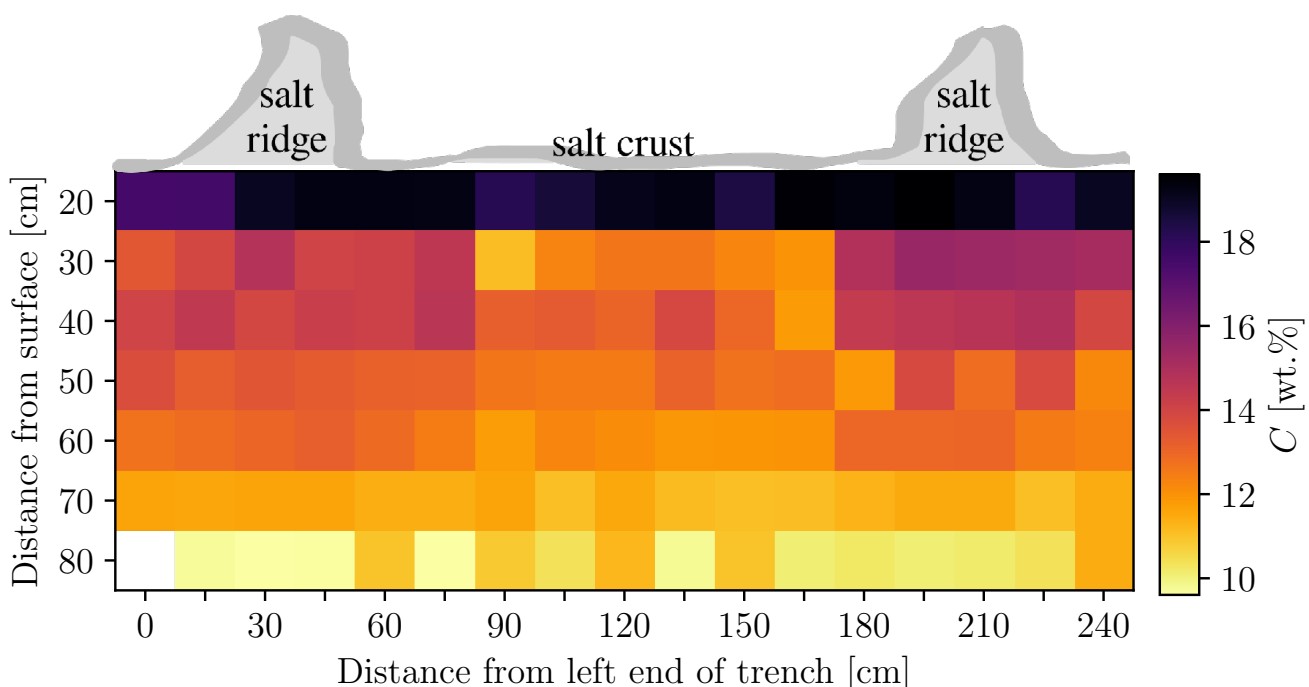

**Figure 7.** Salinity profile compiled from samples taken at site Owens Lake T27-S P1 in January 2018. The color code indicates the salt concentration $C$ in wt.%.



**Figure 8.** Gridded subsets of the surface relief measured by TLS at sites at Owens Lake and Badwater basin. The elevation reflects the surface height above the lowest point in each relief.