# Peer review of "Surface and subsurface characterisation of salt pans expressing polygonal patterns"

_Earth System Science Data, 2020_

## Referee Comment (RC1) · Honghe Xu (Referee) · 6 Jun 2020

This is a data description manuscript on the surface, subsurface and environmental conditions of two salt pans of Owens Lake and Badwater Basin in central California. A big volume of data, including structured measurements and non-structured images and 3-D models, are given in five datasets that are freely accessible through DOI addresses. Authors did lot of works in field campaign and data processing. This is a good job. The data has potential value on further researching of evaporation, formation and evolution of the salt crust and may contribute to the study on evaporates and mineral resources. In this manuscript authors collected data from the two dry lakes. Could authors make comparisons with other related dataset? Also could author state more (a little bit) about the relate research work? I understand this is a data description

manuscript and authors did a lot work. These are just a few advises of mine.

---

## Author Comment (AC1) · 13 Jun 2020

Dear Prof. Hu,

many thanks for your suggestions to improve the manuscript!

Regarding other similar data sets: To my knowledge, there is no data set that combines the types of measurements (temperature & humidity, geochemistry, grain size distributions and TLS surface scans) that we present in our data description. Grain size characterizations are commonly used to characterize the sea floor (see for example (1) https://doi.pangaea.de/10.1594/PANGAEA.746830 and (2) https://doi.pangaea.de/10.1594/PANGAEA.728741). Regarding arid regions, there are a few data sets containing grain size

distributions ((3) https://doi.pangaea.de/10.1594/PANGAEA.913754, (4) https://doi.pangaea.de/10.1594/PANGAEA.871173 and (5) https://doi.pangaea.de/10.1594/PANGAEA.736624) and one other data set that combines a characterization of both the grain size distribution and the geochemistry ( (6) https://doi.pangaea.de/10.1594/PANGAEA.906582). Terrestrial Laser Scan (TLS) data sets are published for example at https://tls.unavco.org/projects/, with one data set originating from the Death Valley - one of our field sites - (7) https://tls.unavco.org/projects/U-062/, which focuses on larger topographic features. We will add these references to other data sets that could be of interest for a potential reader to the manuscript.

Regarding the research context in which the presented data sets have been collected: the goal of the research we conducted was to uncover the mechanism that drives the formation of salt polygons in Salt Playa (see preprints (8) and (9)). These polygons emerge in a range of Salt Playas across the globe, for example the Death Valley, Salar de Uyuni or at the Dead Sea. So far, the driving mechanism is debated and theories that were brought forward to explain the emergence of these patterns are wrinkling (10, 11, 12) an cracking (13, 14, 15, 16, 17). Both of these mechanisms focus on mechanisms that involve only the salt crust in the pattern formation process. We claim that these previously proposed mechanisms are not able to explain both the length scale of the observed patterns (which is on the order of meters) as well as the consistency with which this length scale is expressed at Salt Playas with otherwise very different environmental parameters. As a solution to this problem we propose a third mechanism that includes dynamics of the subsurface of the Salt Playa into account. It has been known for some time that salt lakes can express salinity driven convective dynamics (18) and these convective dynamics have already been shown to occur in the field (19). We propose that the polygonal salt ridges that are visible on the surface grow at the boundary of the convection cells in the underground, as the salt concentration there is higher and supports increased salt precipitation. To support this claim, we characterized both the surface (TLS scans, temperature and humidity measurements)

[Figure]

and the subsurface (grain size distributions, geochemistry, salinity distributions) of two Salt Playas, namely Owens Lake and Badwater Basin in Central California. These characterizations are described in larger detail in the present data publication. We find evidence that supports our claims that convection is the driving mechanism of pattern formation in this region.

As these claims are subject of two separate publications that are in the process of being published in other scientific journals, we would refrain from including a much longer description of the research purpose in the data publication. We can, nevertheless, include a few more details.

References

(1) Michel, Julien; Westphal, Hildegard; Hanebuth, Till J J (2009): (Table 1) Silt grain-size analysis of sediment surface samples in the Golfe d'Arguin. PANGAEA, https://doi.org/10.1594/PANGAEA.746830, Supplement to: Michel, J et al. (2009): Sediment partitioning and winnowing in a mixed eolian-marine system (Mauritanian shelf). Geo-Marine Letters, 29(4), 221-232, https://doi.org/10.1007/s00367-009-0136-8

(2) Sirocko, Frank; Garbe-Schönberg, Carl-Dieter; Devey, Colin W (2000): Composition of sediments from the Arabian Sea. PANGAEA, https://doi.org/10.1594/PANGAEA.728741, Supplement to: Sirocko, F et al. (2000): Processes controlling trace element geochemistry of Arabian Sea sediments during the last 25,000 years. Global and Planetary Change, 26(1-3), 217-303, https://doi.org/10.1016/S0921-8181(00)00046-1

(3) Nottebaum, Veit; Stauch, Georg; van der Wal, Jorien L N; Zander, Anja; Reicherter, Klaus; Batkhishig, Ochirbat; Lehmkuhl, Frank (2020): Grain size and luminescence data from the Orog Nuur Basin (Mongolia). PANGAEA, https://doi.pangaea.de/10.1594/PANGAEA.913754 (dataset in review)

(4) Mischke, Steffen; Liu, Chenglin; Zhang, Chengjun; Zhang, Hua; Jiao, Pengc heng; Plessen, Birgit (2017): Stable oxygen isotope record and grain size distribution of a sediment section in the Tarim Basin. PANGAEA, https://doi.org/10.1594/PANGAEA.871173, Supplement to: Mischke, S et al. (2017): The world's earliest Aral-Sea type disaster: the decline of the Loulan Kingdom in the Tarim Basin. Scientific Reports, 7, 43102, https://doi.org/10.1038/srep43102

(5) Arz, Helge Wolfgang; Lamy, Frank; Pätzold, Jürgen; Müller, Peter J; Prins, Maarten A (2003): Age determination and clay content of sediment core GeoB5804-4. PANGAEA, https://doi.org/10.1594/PANGAEA.736624, Supplement to: Arz, HW et al. (2003): Mediterranean Moisture Source for an Early-Holocene Humid Period in the Northern Red Sea. Science, 300(5616), 118-121, https://doi.org/10.1126/science.1080325

(6) Schwamborn, Georg; Hartmann, Kai; Wünnemann, Bernd; Rösler, Wolfgang; Wefer-Roehl, Annette; Pross, Jörg; Diekmann, Bernhard (2019): Sedimentology, geochemistry and mineralogy of sediment core GN200 from the Gaxun Nur basin (Ejina basin), NW China. PANGAEA, https://doi.org/10.1594/PANGAEA.906582, In: Schwamborn, G et al. (2019): GN200 sediment core from the Gaxun Nur basin (Ejina basin), NW China. PANGAEA, https://doi.org/10.1594/PANGAEA.907462

(7) Terry Pavlis, Mapping Techniques for Metamorphic Terranes, UNAVECO https://tls.unavco.org/projects/U-062/, (20014)

(8) J Lasser et al., Salt Polygons are Caused by Convection, arXiv https://arxiv.org/abs/1902.03600v2 (2020)

(9) M Ernst, J Lasser and L goehring, Stability of convection in dry salt lakes, arXiv https://arxiv.org/abs/2004.10578 (2020)

(10) TK Lowenstein, LA Hardie, Criteria for the recognition of salt-pan evaporites. Sedimentology 32, 627–644 (1985).
(11) FW Christiansen, Polygonal fracture and fold systems in the salt crust, Great Salt Lake Desert, Utah. Science139, 607–609 (1963).

(12) SG Fryberger, AM Al-Sari, TJ Clisham, Eolian Dune, Interdune, Sand Sheet, and Siliciclastic Sabkha Sediments of an Offshore Prograding Sand Sea, Dhahran Area, Saudi Arabia. AAPG Bull.67, 280–312 (1983).

(13) S Lokier, Development and evolution of subaerial halite crust morphologies in a coastal Sabkha setting.J. Arid Environ. 79, 32 – 47 (2012).

(14) D Krinsley, A geomorphological and paleoclimatological study of the playas of Iran. Part 1.U.S. Geol. Surv.CP 70-800(1970).

(15) JC Dixon,Aridic Soils, Patterned Ground, and Desert Pavements. (Springer Netherlands, Dordrecht), pp. 101–122 (2009)

(16) RM Tucker, Giant polygons in the Triassic salt of Cheshire, England; a thermal contraction4 model for their origin.J. Sediment. Res.51, 779 (1981)

(17) PD Deckker, Biological and sedimentary facies of australian salt lakes.Palaeogeogr. Palaeocl.62, 237–270 (1988).

(18) RA Wooding, SW Tyler, I White, PA Anderson, Convection in groundwater below an evaporating Salt Lake: 2. Evolution of fingers or plumes.Water Resour. Res.33, 1219–1228 (1997).

(19) WE Sanford, WW Wood, Hydrology of the coastal sabkhas of abu dhabi, united arab emirates. Hydrogeol. J.9, 358–366 (2001).
* * *

---

## Referee Comment (RC2) · Kevin Perry (Referee) · 25 Aug 2020

General Comments:

The data description paper titled "Surface and subsurface characterisation of salt pans expressing polygonal patterns" provides a comprehensive and detailed summary of a comprehensive suite of measurements collected from the Owens (dry) Lake and Badwater Basin salt pans. The instrumentation, methods, and materials used during the generation of this unique data set are described in exquisite detail. The usefulness of the data set derives from the breadth of surface and subsurface measurements which include both vertical and horizontal cross sections of relevant parameters. Obvious care was taken to identify and avoid potential contamination pathways. In addition,

quality control procedures were implemented, when feasible, to further screen the data to remove potentially contaminated data. The paper was well-written and presented in a logical manner.

Specific Comments:

Data Set: Temperature and humidity time-series from Owens Lake, central California, measured during one week in November 2016. 1) The data at the beginning and end of several of the files are not realistic and should be truncated prior to publication. These data points clearly represent a transient signal prior to equilibration of the sensor. 2) T36-3_P1(center) and T36-3-P1(ridge) were collected using two different sensors with different capabilities (T and RH vs T only) and time resolutions (2 minutes vs 1 minute). These differences make direct comparison of the data sets difficult. Were the two sensors ever tested side by side?

Data Set: TLS surface scans from Owens Lake and Badwater Basin, central California, measured in 2016 and 2018 1) More information on how to extract and display the data from the TLS scan would be useful.

Data Set: Subsurface salt concentration profiles and pore water density measurements from Owens Lake, central California, measured in 2018 1) Owens_lake_T32-1-L1_P3_salt-conc has some negative values for the direct salt content measurements. In addition, the $R^2$ value for the comparison of the direct and indirect measurements of salt content is quite low (i.e., < 0.06). 2) OwensLake-T27-S_salt-conc also has some negative values for the direct salt content measurements. However, the $R^2$ value for the comparison of the direct and indirect measurements of the salt content would be quite good without including these values. 3) OwensLake_porewater includes some standard deviation estimates for data with only 1 or 2 values. Is the stdev measure meaningful in this context? 4) OwensLake_porewater includes some "validation" samples for T16. What exactly are these samples?

Data Set: Chemical characterization of salt samples from Owens Lake and Badwater

Basin, central California, collected in 2016 and 2018 1) There is a discrepancy on the units described in Table 7 for the pore water ion characterization and those contained in the data file OwensLake_porewater-chem. Table 7 indicates that all elements are reported in ug/L while some of the data in the file are reported as mg/L 2) What are the estimated uncertainties of the ICP-OES measurements? 3) Two extraneous rows of empty data are contained at the bottom of OwensLake_crustsaltspecs 4) The abstract for this data set indicates "Pore water was collected from the subsurface at two sites at Badwater Basin (also central California) and 15 sites at Owens Lake. Elemental composition of the pore water was analyzed using ICPOES." However, the data file OwensLake_porewater-chem only contains data for 12 sites. What happened to the data from the other 5 sites?

Technical Comments: 1) Page 19 Line 310 – "were" should be "where" 2) Page 19 Line 312 – "two samples rows" should be "two sample rows"

---

## Author Comment (AC2) · 1 Sep 2020

Dear Prof. Perry,

we would like to thank you for your very thorough comments that greatly helped to improve both the publication as well as the published data sets. It is indeed very helpful to have a second set of eyes comb through the data and help spot all the little mistakes one makes when putting together such a data set.

Following your comments, we have introduced the following additions and changes to the manuscript:

- We have added a sentence describing the repeatability and calibration error of the ICP-OES measurements: "Repeatability of the ICP-OES measurements is about 1%,

with a calibration error of $< 5\%$. Therefore we assume a measurement precision of 5%. Since we did not measure any reference materials, we cannot make statements about the trueness of the measurements."

- We have added a sentence giving more details about the sensors used for the temperature and humidity measurements: "For temperature and relative humidity measurements, we used `HiTemp140` and `RHTemp1000IS` data loggers, which measured temperatures and relative humidity every $\Delta t = 60$ seconds or $\Delta t = 120$ seconds with a precision of $\pm 0.01\,°C$ and $\pm 0.1\,\%$, respectively. The factory calibration was used for all sensors. Unfortunately, we did not test sensors side-by-side but values from different sensors seemed consistent."

- We have added a sentence explaining the purpose of the "validation" samples for the pore water density measurements: "Individual measurements of densities are reported, along with averages and standard deviations for all measurements on individual samples. In addition to several replication measurements of the same sample, the data set also contains 5 "validation" samples for site T16 P1. These samples were taken in close spatial proximity to their respective counterparts (indicated by their shared sample ID) and represent replications on the sample level."

- We added a sentence pointing out that we truncated the temperature and humidity data in the beginning and end to get rid of transients: "We truncated the recorded data at the beginning and end to remove data points corresponding to a transient phase directly after putting the sensors in place and after removing them, respectively."

- We added an explanation on how the raw and gridded TLS data files can be loaded: "Raw and gridded data is stored as space-separated `.txt` and `.xyz` files and can be read for example using the `numpy.loadtxt()` in Python."

- We added a paragraph, explaining the exclusion of some data points from the salt concentration profile data sets. We also really liked your idea of calculating $R^2$ values for the direct vs. indirect measurements of salt content and incorporated this as a mea-

sure of the agreement between the two approaches: "During the laboratory analysis of the salt contained in the samples, a small number of samples was contaminated or lost due to mistakes in the dilution process or broken crystallisation dishes. Consequently, these data points are missing either from the direct or indirect measurement column in the data set. Agreement between the direct and indirect measurement for all three sites is very high, with $R^2 = 0.98$ ($p < 0.001$) for site T27-S P1, $R^2 = 0.96$ ($p < 0.001$) for site T32-1-L1 P2 and $R^2 = 0.93$ ($p < 0.001$) for site T32-1-L1 P3."

Additionally, we have requested an update to the data files stored at PANGAEA. All requested changes are listed in the attached document.

Please also note the supplement to this comment:
https://essd.copernicus.org/preprints/essd-2020-86/essd-2020-86-AC2-supplement.pdf

―――――――――――――――――――

**Supplement:**

**Data set changes - 2020-09-01**

**Salt concentration data (DOI: 10.1594/PANGAEA.911059)**

**Owens_lake_T32-1-L1_P3_salt-conc**
- replaced 0 with NaN values
- removed (17,1) to (17,9) and (18,1) to (18,9) direct salt measurements. Found lab notes that for these measurements there was a mixup of dishes in the lab
- removed (14,1) to (14,9) and (15,1) to (15,9) direct salt measurements. Did not find lab notes indicating potential mixups or contamination but the results from the direct measurements do not look trustworthy when compared to the indirect measurements.

**OwensLake-T27-S_salt-conc**
- removed all negative values from the direct salt measurement. These should have been missing values (the cristallization dishes were dropped, according to the lab notes) but were wrongly calculated and included

**OwensLake_porewater**
- removed entries from column "measurement_error" for cases with <= 2 measurements
- swapped entries from columns "z" and "x" for site T25-3_P2 and T16 (these were in the wrong order)
- added a depth entry (depth of 0 m) to entries from site T16 (these samples were drawn from water pooling into the bore holes)
- converted entries in column "Dinstance from left ridge" from centimeters to meters
- renamed T-10-3 -> T-10-3_P1, T16 -> T16_P1 and T16_validation -> T16_P1_validation to be consistent with the other data sets

**Salt characterization data (DOI: 10.1594/PANGAEA.911239)**

**OwensLake_crustsaltspecs**
- renamed T2-5 -> T2-5_P1 and T10-3 -> T10-3_P1 to be consistent with the other data sets
- renamed column "Perc [%] (Sodium Hydrogen Carbonate Hyd...)" -> "Perc [%] (Trona, $Na_3(HCO_3)(CO_3)\cdot2H_2O$)"
- renamed column "Perc [%] (Sodium Chloride)" -> "Perc [%] (Halite, NaCl)"
- renamed column "Perc [%] (Sodium Carbonate Sulfate)" -> "Perc [%] (Burkeite, $Na_6(CO_3)(SO_4)_2$)"
- renamed column "Perc [%] (Sodium Sulfate)" -> "Perc [%] (Thenardite, $Na_2SO_4$)"
- renamed column "Perc [%] (Calcium Carbonate)" -> "Perc [%] (Calcite, $CaCO_3$)"
- renamed column "Perc [%] (Sodium Calcium Aluminum Silicate)" -> "Perc [%] (Albite, $NaAlSi_3O_8$)"

- renamed column "Perc [%] (Sodium Hydrogen Carbonate)" -> "Perc [%] (Nahcolite, NaHCO_3)"
- renamed column "Perc [%] (Sodium Calcium Manganese Tita...)" -> "Perc [%] (Janhaugite, (Na,Ca)_3(Mn,Fe)_3(Ti,Zr,Nb)_2Si_4O_{16}(OH,F)_2)"
- renamed column "Perc [%] (Silicon Oxide)" -> "Perc [%] (Quartz, SiO_2)"
- renamed colum "Perc [%] (Potassium Aluminum Silicate)" -> "Perc [%] (Orthoclase, KAlSi_3O_8)"
- renamed column "Perc [%] (Magnesium Calcium Carbonate)" -> "Perc [%] (Siderophyllite, KFe_2Al(Al_2Si_2)O_{10}(F,OH)_2)"
- renamed column "Perc [%] (Magnesium Calcium Carbonate)" -> "Perc [%] (Huntite, Mg_3Ca(CO_3)_4)"
- renamed column "Perc [%] (Calcium Hydrogen Phosphate Hy...)" -> "Perc [%] (CaHPO_4)"
- removed columns "Perc [%] (Strontium Cesium Aluminum Silicate)", "Perc [%] (Antimony Oxide)", "Perc [%] (Magnesium Iron Iron Aluminum )" and "Perc [%] (Calcium Hydroxide)" since these entries only referred to a trace elements
- removed two superfluous rows at the end of the table
- data abstract corrected to read "Pore water was collected from the subsurface at two sites at Badwater Basin (also central California) and 10 sites at Owens Lake."

**Temperature & humidity data (DOI: 10.1594/PANGAEA.911139)**

**OwensLake_T27-A_P1_Temp_center**
- removed 6 entries at the beginning
- removed one entry at the end

**OwensLake_T27-A_P1_Temp_center2**
- removed 6 entries at the beginning
- removed one entry at the end

**OwensLake_T29-3_P1_Temp_center**
- removed 6 entries at the beginning
- removed two entries at the end

**OwensLake_T29-3_P1_TempRH_ridge**
- removed 19 entries at the beginning

**OwensLake_T36-3_P1_TempRH_center**
- removed 23 entries at the beginning

**OwensLake_T36-3_P1_TempRH_ridge**
- removed 47 entries at the beginning

---

## Author Comment (AC3) · 1 Sep 2020

We also corrected the units in table 7 to correctly reflect when $\mu$g and when mg were reported.
* * *

---

## Author Response (AR1)

Dear Prof. Hu,

many thanks for your suggestions to improve the manuscript!

Regarding other similar data sets: To my knowledge, there is no data set that combines the types of measurements (temperature & humidity, geochemistry, grain size distributions and TLS surface scans) that we present in our data description. Grain size characterizations are commonly used to characterize the sea floor (see for example (1) https://doi.pangaea.de/10.1594/PANGAEA.746830 and (2) https://doi.pangaea.de/10.1594/PANGAEA.728741). Regarding arid regions, there are a few data sets containing grain size

distributions ((3) https://doi.pangaea.de/10.1594/PANGAEA.913754, (4) https://doi.pangaea.de/10.1594/PANGAEA.871173 and (5) https://doi.pangaea.de/10.1594/PANGAEA.736624) and one other data set that combines a characterization of both the grain size distribution and the geochemistry ( (6) https://doi.pangaea.de/10.1594/PANGAEA.906582). Terrestrial Laser Scan (TLS) data sets are published for example at https://tls.unavco.org/projects/, with one data set originating from the Death Valley - one of our field sites - (7) https://tls.unavco.org/projects/U-062/, which focuses on larger topographic features. We will add these references to other data sets that could be of interest for a potential reader to the manuscript.

Regarding the research context in which the presented data sets have been collected: the goal of the research we conducted was to uncover the mechanism that drives the formation of salt polygons in Salt Playa (see preprints (8) and (9)). These polygons emerge in a range of Salt Playas across the globe, for example the Death Valley, Salar de Uyuni or at the Dead Sea. So far, the driving mechanism is debated and theories that were brought forward to explain the emergence of these patterns are wrinkling (10, 11, 12) an cracking (13, 14, 15, 16, 17). Both of these mechanisms focus on mechanisms that involve only the salt crust in the pattern formation process. We claim that these previously proposed mechanisms are not able to explain both the length scale of the observed patterns (which is on the order of meters) as well as the consistency with which this length scale is expressed at Salt Playas with otherwise very different environmental parameters. As a solution to this problem we propose a third mechanism that includes dynamics of the subsurface of the Salt Playa into account. It has been known for some time that salt lakes can express salinity driven convective dynamics (18) and these convective dynamics have already been shown to occur in the field (19). We propose that the polygonal salt ridges that are visible on the surface grow at the boundary of the convection cells in the underground, as the salt concentration there is higher and supports increased salt precipitation. To support this claim, we characterized both the surface (TLS scans, temperature and humidity measurements)

[Figure]

and the subsurface (grain size distributions, geochemistry, salinity distributions) of two Salt Playas, namely Owens Lake and Badwater Basin in Central California. These characterizations are described in larger detail in the present data publication. We find evidence that supports our claims that convection is the driving mechanism of pattern formation in this region.

As these claims are subject of two separate publications that are in the process of being published in other scientific journals, we would refrain from including a much longer description of the research purpose in the data publication. We can, nevertheless, include a few more details.

[Figure]

we would like to thank you for your very thorough comments that greatly helped to improve both the publication as well as the published data sets. It is indeed very helpful to have a second set of eyes comb through the data and help spot all the little mistakes one makes when putting together such a data set.

Following your comments, we have introduced the following additions and changes to the manuscript:

- We have added a sentence describing the repeatability and calibration error of the ICP-OES measurements: "Repeatability of the ICP-OES measurements is about 1%,

with a calibration error of $< 5\%$. Therefore we assume a measurement precision of 5%. Since we did not measure any reference materials, we cannot make statements about the trueness of the measurements."

- We have added a sentence giving more details about the sensors used for the temperature and humidity measurements: "For temperature and relative humidity measurements, we used `HiTemp140` and `RHTemp1000IS` data loggers, which measured temperatures and relative humidity every $\Delta t = 60$ seconds or $\Delta t = 120$ seconds with a precision of $\pm 0.01\,°C$ and $\pm 0.1\,\%$, respectively. The factory calibration was used for all sensors. Unfortunately, we did not test sensors side-by-side but values from different sensors seemed consistent."

- We have added a sentence explaining the purpose of the "validation" samples for the pore water density measurements: "Individual measurements of densities are reported, along with averages and standard deviations for all measurements on individual samples. In addition to several replication measurements of the same sample, the data set also contains 5 "validation" samples for site T16 P1. These samples were taken in close spatial proximity to their respective counterparts (indicated by their shared sample ID) and represent replications on the sample level."

- We added a sentence pointing out that we truncated the temperature and humidity data in the beginning and end to get rid of transients: "We truncated the recorded data at the beginning and end to remove data points corresponding to a transient phase directly after putting the sensors in place and after removing them, respectively."

- We added an explanation on how the raw and gridded TLS data files can be loaded: "Raw and gridded data is stored as space-separated `.txt` and `.xyz` files and can be read for example using the `numpy.loadtxt()` in Python."

- We added a paragraph, explaining the exclusion of some data points from the salt concentration profile data sets. We also really liked your idea of calculating $R^2$ values for the direct vs. indirect measurements of salt content and incorporated this as a mea-

sure of the agreement between the two approaches: "During the laboratory analysis of the salt contained in the samples, a small number of samples was contaminated or lost due to mistakes in the dilution process or broken crystallisation dishes. Consequently, these data points are missing either from the direct or indirect measurement column in the data set. Agreement between the direct and indirect measurement for all three sites is very high, with $R^2 = 0.98$ ($p < 0.001$) for site T27-S P1, $R^2 = 0.96$ ($p < 0.001$) for site T32-1-L1 P2 and $R^2 = 0.93$ ($p < 0.001$) for site T32-1-L1 P3."

Additionally, we have requested an update to the data files stored at PANGAEA. All requested changes are listed in the attached document.

Please also note the supplement to this comment:
https://essd.copernicus.org/preprints/essd-2020-86/essd-2020-86-AC2-supplement.pdf

[Figure]

**Surface and subsurface characterisation of salt pans expressing polygonal patterns**

Jana Lasser[1,2], Joanna M. Nield[3], and Lucas Goehring[4]

[1]Max Planck Institute for Dynamics and Self-Organization, Am Fassberg 17, 37077 Göttingen, Germany
[2]Complexity Science Hub Vienna, Josefstädterstrasse 39, 1080 Wien
[3]Geography and Environmental Science, University of Southampton, Highfield, Southampton SO17 1BJ, UK
[4]School of Science and Technology, Nottingham Trent University, Nottingham NG11 8NS, UK

**Correspondence:** Jana Lasser (lasser@csh.ac.at)

**Abstract.** The data set described here contains information about the surface, subsurface and environmental conditions of salt pans that express polygonal patterns in their surface salt crust (Lasser et al., 2020b), DOI: 10.5880/fidgeo.2020.037. Information stems from 5 field sites at Badwater Basin and 21 field sites at Owens Lake – both in central California. All data was recorded during two field campaigns, from between November and December, 2016, and in January 2018. Crust surfaces, including the mean diameter and fluctuations in the height of the polygonal patterns, were characterised by terrestrial laser scanner. The data contains the resulting three dimensional point clouds, which describe these surfaces. The subsurface is characterised by grain size distributions of samples taken from depths between 5 cm and 100 cm below the salt crust, and measured with a laser particle size analyser. Subsurface salinity profiles were recorded and the ground water density was also measured. Additionally, the salts present in the crust and pore water were analysed to determine their composition. To characterise the environmental conditions at Owens Lake, including the differences between nearby crust features, records were made of the temperature and relative humidity during one week in November 2016. The field sites are characterised by images, showing the general context of each site, such as pictures of selected salt polygons, including any which were sampled, a typical core from each site at which core samples were taken and close-ups of the salt crust morphology. Finally, two videos of salt crust growth over the course of spring 2018 and reconstructed from time-lapse images are included.

*Copyright statement.* The data sets referenced in this publication are made available under the Attribution International 4.0 license (CC-BY 4.0).

[revised manuscript text omitted]

---

## Author Response (AR2)

Liebe Kirsten,

Alle DOIs sind jetzt registriert (habe ich überprüft) und alle links zu DOIs im Paper sind jetzt entsprechend angepasst und führen zu doi.org/…

Liebe Grüße,
Jana